# Gold Nanodisks Plasmonic Array for Hydrogen Sensing at Low Temperature

**DOI:** 10.3390/s19030647

**Published:** 2019-02-05

**Authors:** Marco Sturaro, Gabriele Zacco, Pierfrancesco Zilio, Alessandro Surpi, Marco Bazzan, Alessandro Martucci

**Affiliations:** 1Dipartimento di Ingegneria Industriale, Università di Padova, 35131 Padova, Italy; sturarom@gmail.com; 2Veneto Nanotech, 35100 Padova, Italy; gabriele.zacco@al-lighting.com (G.Z.); pierfrancesco.zilio@gmail.com (P.Z.); asurpi@bo.ismn.cnr.it (A.S.); 3Dipartimento di Fisica e Astronomia, Università di Padova, 35131 Padova, Italy; marco.bazzan@unipd.it

**Keywords:** plasmonic, gold array, optical gas sensors, hydrogen sensors, nanoimprinting

## Abstract

We present a novel plasmonic hydrogen sensor consisting of an array of gold nanodisks produced by lithography. The size, height, and spacing of the disks were optimized using finite element simulation to generate a sharp localized surface plasmon resonance peak in the near-infrared wavelength region. The reported results show the possibility of developing an optical gas sensors-based bare Au nanostructures operating at a low temperature.

## 1. Introduction

Gold is regarded as relatively inert to chemical interactions with gases and generally considered a poor catalyst. Instead, gold nanoclusters up to 5 nm of diameter show catalytic activity to CO and H_2_ oxidation [1,2], especially when supported by transition metal oxides [3]. The reactivity of small Au nanocrystals (NCs) is largely attributed to the presence of low-coordinated Au atoms, abundant in nanometer-sized nanoparticles (NPs) [4]. Furthermore, Density Functional Theory (DFT) simulations confirm that the crucial condition for H_2_ interaction with gold is the existence of low-coordinated atoms, independently if on the surface of Au NCs or at extended line defects [5]. In the latter case, it is possible to observe significant catalytic activity in larger gold structures. For example, Fujita et al. [6] studied CO oxidation activity in nanoporous gold, where the feature characteristic lengths were larger than 30 nm but with peculiar defective surfaces comparable to 3–5 nm Au NCs curvature. Recently, bare gold nanostructures have been exploited to obtain hydrogen sensors at low temperature. The proposed mechanism involves hot electron-induced hydrogen molecule dissociation and subsequent metastable hydride formation [7,8]. 

In this paper, we study the interactions of H_2_ with a metamaterial consisting of bare gold nanodisks (Au NDs) arrays on silica substrate. The aim was to verify the possibility of optically detecting hydrogen at low temperature using only gold as the sensitive material without any particular excitations and exploiting the shift in the localized surface plasmon resonance (LSPR) of the gold array.

## 2. Materials and Methods

### 2.1. Fabrication Method

A master was realized on silicon substrate by means of Lloyd Interferential Lithography (LIL) and inductively coupled plasma (ICP) etching. In the second phase, the master on silicon was replicated numerous times on quartz substrates. The last phase of the process comprised metallization of chromium and gold by electron beam evaporation and a subsequent lift-off. 

The master mold was produced by an interferential lithography system in a “Lloyd’s Mirror” configuration. In this configuration, the interference pattern is created by the superposition of a portion of a laser beam directly reaching the sample and another portion of the beam reflected by a mirror placed at 90° to the sample. The light source used was a 50 mW HeCd laser emitting a single mode at λ = 325 nm and 30 cm coherence length. The laser beam was directed on a spatial filter to eliminate spurious frequencies and then continued free to propagate, achieving a distance of 2 m and reaching the Lloyd interferometer. 

Bottom Anti-Reflective Coating (BARC) was deposited onto silicon by spin coating at 3000 rpm for 30 s and annealed on a hot plate at 175 °C for 1 min. Photoresist was a SPR 220 diluted in propylene glycol methyl ether acetate (PGMEA), deposited by spin coating at 5000 rpm for 30 s and placed on a hot plate for 90 s. 

Sample obtained after lithography constituted NDs of photoresist on BARC/silicon. Subsequently, a transfer process was performed using ICP etching. The master obtained, shown in Appendix A, was then used as a mold for nanoimprinting. Quartz slides (6 cm^2^) were cleaned in ultrasonic bath and washed for several cycles with water, soap, and isopropyl alcohol. Next, mr-I 7010 R thermoplastic polymer—Microresist (MRI), was deposited on quartz by spin coating (1500 rpm for 30 s and annealed at 100 °C for 1 min). Quartz slide and mold were then joined at a pressure of 100 bar and 100 °C for 10 min. Finally, chromium and gold evaporation by electron beam was performed.

### 2.2. Characterizations

The morphology of the NDs array was investigated with a xT Nova NanoLab Scanning Electron Microscopy (SEM).

X-ray diffraction (XRD) using the CuK_α_ (λ = 1.54056 Å) radiation was performed using a Philips MRD diffractometer, equipped with a parabolic multilayer mirror for primary beam conditioning and an Anton-Paar sample chamber which allowed investigation of the structural characteristics of the samples in a controlled atmosphere and temperature. The diffracted beam was measured as a function of the scattering angle by a proportional Xe counter coupled to a Parallel Plate Collimator with an angular acceptance of 0.01°.

Transmittance at normal incidence and ellipsometry quantities Ψ and Δ were measured using a J.A. Woollam V-VASE Spectroscopic Ellipsometer in vertical configuration at a fixed angle of 70° in the wavelength range 300–1500 nm. Refractive index n and extinction coefficient k have been evaluated from Ψ, Δ, and transmittance data using the WVASE32 ellipsometry data analysis software fitting the experimental data with Cauchy dispersion and Gaussian oscillators. A cell that allows heating and introduction of gases (nitrogen and a mixture of 5 vol% of H_2_ in Ar at 0.4 L/min) was used to observe changes in sample optical constants in the presence of hydrogen.

Optical gas sensing tests were performed via optical absorption measurements over the wavelength range 300 nm < *λ* < 2000 nm using a Harrick gas flow cell coupled with a Jasco V-570 spectrophotometer. NDs arrays deposited on SiO_2_ substrates were exposed at different operative temperatures (from 30 to 300 °C) to 1% v/v H_2_ balanced in dry air, at a flow rate of 0.4 L/min. All tested samples were stabilized at the operative temperature for one hour in air before gas sensing measurements. Sensor sensitivity was assessed using response intensity (RI) defined as RI = ǀ1−Abs_Gas_/Abs_Air_ǀ. To analyze the dynamic behavior of the sensor, we used the response and recovery time calculated as the time needed to reach 90% of the total response and to recover 90% of the baseline.

## 3. Results and Discussion

LSPR absorption peak can be effectively tuned by changing the aspect ratio of Au NDs and the periodicity of the array [9]. By using COMSOL Multiphysics, a commercial software which uses finite element method (FEM), the optical response of a single gold ND in air having glass as substrate was modelled, and its radius and height were optimized to obtain a strong and sharp LSPR peak. Subsequently, the procedure was repeated for NDs square arrays to optimize ND spacing. Simulations (see Appendix A) indicate that increasing the ND diameter leads to a red shift of the LSPR peak position [9], passing from λ_LSPR_ = 700 nm (visible range) for a ND with radius of 25 nm to λ_LSPR_ = 1100 nm (NIR) for a ND with radius of 150 nm. ND thickness increment, for every simulated width, had the opposite effect on the LSPR peak position (i.e., increasing thickness induced a blue shift of the LSPR peak position); 5 nm thickness was chosen for subsequent simulations because the NDs have the sharpest LSPR in the visible/NIR range for that thickness. Similar absorption cross sections for NDs arrays with period length between 250 and 450 nm at λ_LSPR_ ~ 1100 nm were obtained with successive simulations (Figure 1). On the basis of simulations, samples containing NDs with 75 nm radius and 5 nm thickness with a periodicity of 400 nm were fabricated by nanoimprinting from a master mold produced by interferential lithography (Figure 1c) [10]. Using this approach, a large number of samples can be produced from a single master while maintaining high resolution and high experimental reproducibility. 

The samples studied for H_2_ sensing were arrays of gold NDs on quartz, each with a radius of 75 nm and height 5 nm with a period of 400 nm. The absorption spectra is shown in Figure 2, in which the LSPR is centered at ~1100 nm, as predicted by simulations. SEM images of the sample are presented in Figure 2, showing well-defined, long-range order and quite regular NDs shapes.

The sensing behavior of Au NDs was investigated using H_2_ (1 vol% in air) as target gas and dry air as reference gas. The samples were stabilized in air at the operative temperature for 1 hour before gas sensing measurements. Stabilization is important because the temperature affects the size of the NDs and hence the position of the LSPR, as reported in Appendix A. The shift of the LSPR peak observed in Appendix A is consistent with the simulations, which predict a blue shift of the LSPR peak as the diameter of the Au ND decreases (Appendix A).

As seen in Figure 3, the LSPR peak shifts to lower wavelengths, passing from air to H_2_ at different operative temperature; this effect is reversible. The effect of H_2_ exposure can be better appreciated by plotting the difference of the absorbance and the wavelength (Optical Absorbance Change, OAC=Abs_H_2__–Abs_Air_), as reported in Figure 4. The response is wavelength dependent and influenced by the operative temperature, even if a detectable signal also can be obtained at 30 °C. Figure 4a shows that the OAC has a maximum at ~950 nm (for all operative temperatures) and a minimum (dependent on operative temperatures) in the 1100–1400 nm range, which can be used for dynamic tests. Figure 4b,c show the dynamic tests at 950 nm and for two operative temperatures (100 °C and 150 °C) for two air/H_2_/air cycles. The sensor exhibits better performance at higher operative temperatures in terms of response, recovery time, and response intensity (RI), which were estimated (using the data reported in Figure 4c) to be 100 s, 140 s, and 0.1%, respectively. At temperatures above 200 °C, sensitivity declines and the LSPR peak tends to decrease in intensity as a result of the gradual change of the Au NDs dimension, as reported in Appendix A. These experiments demonstrate that our bare Au NDs array can detect hydrogen at relatively low temperature, a phenomenon observed before for bare Au only in cases of very small gold clusters [5] or involving hot electrons [7,8].

The observed LSPR shift could be related to a variation of the refractive index. However, the shift due to refractive index variation (hydrogen refractive index is slightly lower than air) is negligible if compared with the shift observed experimentally. Moreover, ellipsometry measurements showed no variations due to gas exposure (see Appendix A). 

Another possible mechanism involves the formation of gold-hydrogen bonds [5], preferentially at the disks’ edges, where gold atoms are low coordinated. We performed XRD measurements (see Appendix A) to investigate the influence of hydrogen interaction with the Au NDs, but the XRD diffraction peak of Au was not affected by the presence of hydrogen. This experiment does not exclude the hydrogen–gold bond formation because the hydrogen dissociation on gold would occur preferentially only on the surfaces edges, which is difficult to detect by standard XRD. Nevertheless, XRD measurements provided important information about Au crystallite size, calculated using the Scherrer equation. Au NDs are polycrystalline structures with a mean crystallite diameter of 6 ± 0.5 nm, the same order of magnitude of an Au cluster presenting activity to hydrogen dissociation [1,2,5]. However, direct hydrogen dissociation at low temperature has been demonstrated only for small Au clusters, supported on metal oxides such as TiO_2_ and CeO_2_ [11,12].

In our experiments, we did not use any type of light excitation source for exciting hot electrons as reported in [7,8,13,14], obtaining a detectable shift also at 30 °C. Therefore, the formation of hot electrons seems to be not very likely to occur.

## 4. Conclusions

In summary, we present a new sensing platform for hydrogen on the basis of the enhanced sensing performance of bare gold resulting from the sharp plasmon resonance of a specifically designed nanodisks pattern fabricated by nanoimprinting. The Au NDs array showed a reversible response to hydrogen at low temperature with a response intensity of 0.1% and a response time of 100 s at an operative temperature of 150 °C. Although the mechanisms involved in the observed LSPR shift is not completely understood, we demonstrated the possibility to design a plasmonic structure made of bare gold with good sensitivity and reversibility for H_2_ sensing.

## Figures and Tables

**Figure 1 sensors-19-00647-f001:**
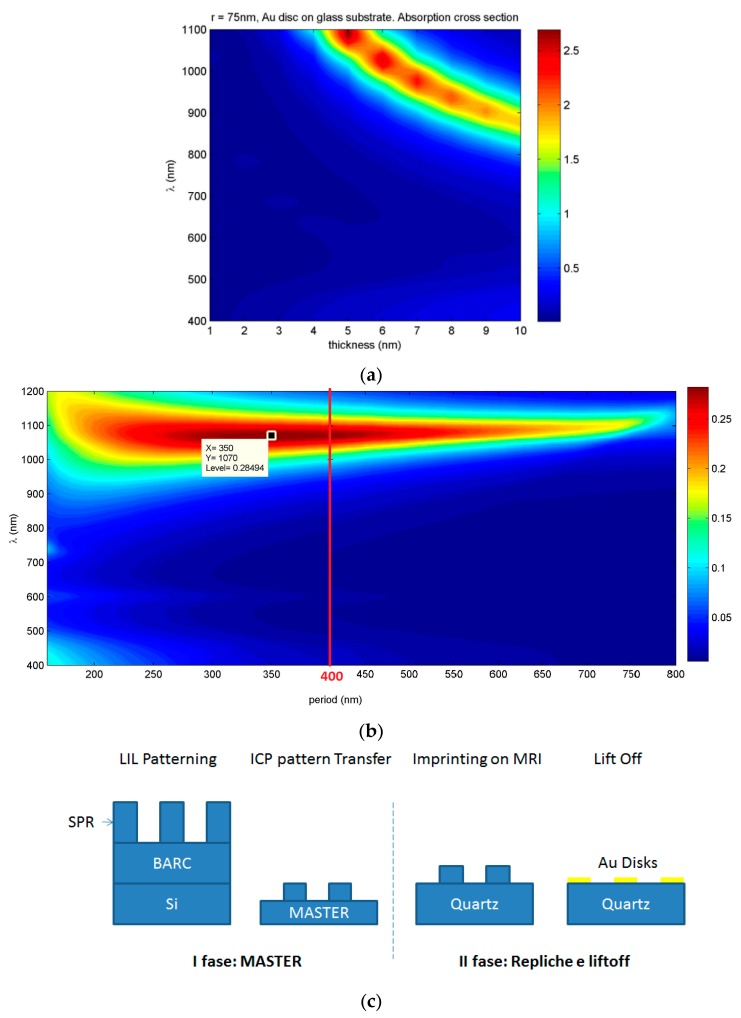
(**a**) Absorption cross-section of single Au ND as a function of wavelength and ND thickness for radius = 75 nm; (**b**) Absorption cross section of Au NDs array as a function of wavelength and periodicity for radius = 75 nm, thickness = 5 nm; (**c**) Scheme of sample preparation method.

**Figure 2 sensors-19-00647-f002:**
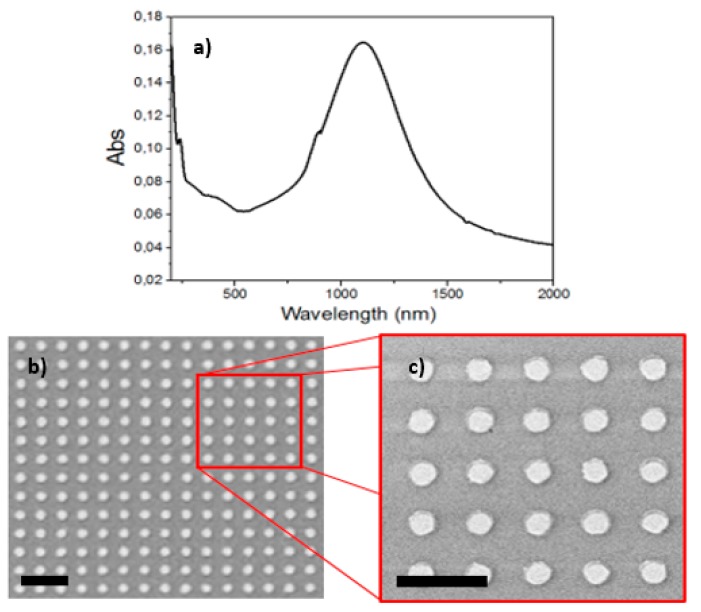
(**a**) Optical absorption spectrum in the 200–2000 nm range and (**b**,**c**) SEM images of Au NDs array on quartz with diameter of 150 nm and height of 5 nm. Scale bars are 500 nm.

**Figure 3 sensors-19-00647-f003:**
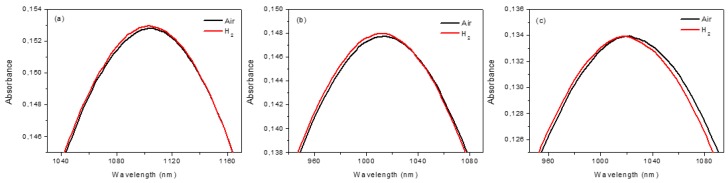
Optical absorption spectra for Au NDs at (**a**) 30 °C, (**b**) 100 °C, and (**c**) 150 °C operative temperature in air (black line) and in H_2_ (red line).

**Figure 4 sensors-19-00647-f004:**
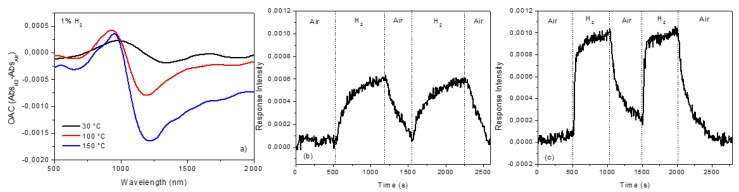
(**a**) Optical Absorbance Change, OAC=Abs_H_2__-Abs_Air_**,** for Au NDs at 30 °C, 100 °C, and 150 °C operative temperature when exposed to 1% H_2_ in air. Dynamic cycle air/H_2_/air at 950 nm and 100 °C (**b**) and 150 °C (**c**) operative temperature.

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
