# Peer review of "Gold Nanodisks Plasmonic Array for Hydrogen Sensing at Low Temperature"

_sensors, 2019, doi:10.3390/s19030647_

Round 1
Reviewer 1 Report
This paper proposes a new hydrogen sensing platform based on the localized surface plasmon resonance (LSPR) of an array of gold nanodisks (Au NDs). The authors theoretically explored the optimized size, height and spacing of gold nanodisks with the LSPR peak in the near infrared region and fabricated the gold nanodisks by lithographic methods. At different temperatures, both the absorbance and LSPR peak of gold nanodisks changed when exposed in 1% H2. The experimental results show a good sensitivity and reversibility for H2 sensing at low temperature. However, the reason of the LSPR sensing mechanism is not clear. I do not think the current manuscript can be published. It may deserve publication after a major revision addressing some key issues. [1] Could the author provide the peak shift of the gold nanodisk LSPR caused by the refractive index change? [2] The authors performed some experiments at high temperatures, at which the gold will suffer from surface melting, instead of degradation. At a temperature of 150 degree C, the gold nanodisk can be annealed to form larger crystallite size. Could the author provide the data of gold crystallite size at different temperatures, since larger crystalline size at high temperature may lead to poorer H2 adsorption and therefore worse sensing performance? [3] The authors performed optical measurements, which unavoidably will excite the LSPR of gold nanodisks. However, the authors claimed that the possible H2 adsorption is not caused by the LSPR-generated hot electrons. [4] Please provide the optical absorption spectra of Au NDs in the 300 - 2000 nm range at different temperature (30, 100, and 150°C), corresponding to the optical absorbance change plot shown in Figure 4a. [5] The author claimed that at temperatures above 200°C, the sensitivity worsens and the LSPR peak tends to decrease in intensity, due to the gradual degradation of the NDs structures. But the experimental results are not given in the article. [6] The hydrogen sensing at low temperature was performed by the measurement of optical absorbance change. Figure 4b and 4c show the dynamic hydrogen sensing at 950 nm and for two operative temperatures (100°C and 150°C). But it can be seen that the optical absorbance change was more obvious in the range of 1000 - 1500 nm as shown in Figure 4a. Please explain why used the wavelength at 950 nm rather than in the range of 1000 - 1500 nm. [7] There are some grammar errors, for example, 'read shift' should be 'red shift'.Author Response
All the raised points have been addressed in the attached file. We thank the reviewer for the very useful suggestions and we hope that the revised version could be accepted for publication.

Reviewer 2 Report
Comments are in attached documents. Without the proof of hydride formation this paper cannot be accepted
More references need to be added to the paper

Author Response
All the raised points have been addressed in the attached file. We thank the reviewer for the very useful suggestions and we hope that the revised version could be accepted for publication.

Reviewer 3 Report
Marco Sturaro et al. present in their manuscript a numerical and experimental study for hydrogen sensing made of a plasmonic nanoantenna. The plasmonic nanoantenna is made of gold nanorods whose appropriate numerical investigation leads to high quality factor of the localised surface plasmon polariton supported by the single nanoantennas in the array. The measured sensor has an appreciable absorbance change of the peak resonance when 1% H2 is substituted to the reference medium at a temperature of 150 C, namely air.
Author Response

(The authors gave the same response as above.)

Round 2
Reviewer 1 Report
I think the reply of the authors to the referee's comments is good. The authors have revised the manuscript by adding enough details. I thus recommend the publication of this article.
Author Response
We thank the reviewer for the positive response.
Reviewer 2 Report
NA
Author Response
We thank the reviewer for the suggestions. The paper has been revised improving the description of the experiments and the results. All the changes are marked in yellow in the revised manuscript.
Reviewer 3 Report
The authors have greatly improved their manuscript and the level of presentation is clear and appropriate for the standards of Sensors.
However, it does not seem the authors have addressed the main criticism I have raised in my first review:
1) the authors do not give a full explanation for the change in the absorbance, they conclude that it cannot depend on a variation of the refractive index, but they lack a full evidence of the second hypothesis, i.e. the formation of gold-hydrogen bonds at the nanorod surface. For this reason, at the current stage, the manuscript is not ready for publication in Sensors.
In this regard, I believe that a surface spectroscopy as for example, X-ray photoelectron spectroscopy, could perhaps confirm the formation of gold-hydrogen bonds.
Author Response
We agree that we did not prove the formation of gold hydrogen bond, but the observed LSPR shift is evident at 150 °C, so it is not possible to perform XPS at such high temperature at least with the XPS facilities at which we can have access. On the other hand, also FTIR or Raman at high temperature are not easy to perform, considering that our sample are deposited on glass and FTIR could be done only in ATR configuration. We would like to point out that we tried in situ XRD experiments at 150°C and we did not see any variation on the Au XRD peaks. Therefore, even if it is not possible to exclude the formation of a very thin superficial layer of hydrogen gold bond, for sure we can exclude variation of the “bulk” lattice of the Au crystals.
As stated in the paper, the main aim was verifying the possibility to optically detect hydrogen at low temperature, using only gold as sensitive material without any particular excitations and, to best of our knowledge, this effect has not been previously reported, at least for bare Au nanodisks array.